# How Effective Is Environmental Protection for Ensuring the Vitality of Wild Orchid Species? A Case Study of a Protected Area in Italy

**DOI:** 10.3390/plants13050610

**Published:** 2024-02-23

**Authors:** Lisa Scramoncin, Renato Gerdol, Lisa Brancaleoni

**Affiliations:** Department of Environmental and Prevention Sciences, University of Ferrara, C.so Ercole I D’Este 32, 44121 Ferrara, Italy; lisa.scramoncin@unife.it (L.S.); bcl@unife.it (L.B.)

**Keywords:** environmental threats, Natura 2000, population vitality, protected areas, reproductive traits, vegetative traits

## Abstract

Orchids are among the plants most threatened by anthropic impact and environmental changes. Therefore, all known orchid species are protected in several countries by regional, national and international legislation. Several studies have cast doubts on the effectiveness of legislation to ensure the protection of wild orchids. We assessed the vitality of four orchid populations in a coastal area in Northern Italy, by monitoring the vegetative and reproductive traits of the orchid populations growing both in the protected sites comprising the Natura 2000 network, and in non-protected sites. We also monitored the level of environmental threat to orchid vitality. The early-flowering deceptive species (*Ophrys sphegodes* and *Anacamptis morio*) exhibited high vegetative vitality and experienced similar levels of environmental threat in the protected and non-protected areas. However, their reproductive success was strongly jeopardized, probably by a failed pollination that impeded the fruit set. The late-flowering, partially or totally rewarding species (*Anacamptis pyramidalis* and *Anacamptis coriophora*) were more strongly impacted by spring mowing and ungulate herbivory and alien species. Only for *A. coriophora* were the herbivory and alien species invasions lower at the protected vs. non-protected sites, which ensured a higher population vitality at the protected sites. We conclude that the environmental protection in our study area is ineffective for preserving orchids without targeted actions against specific environmental threats.

## 1. Introduction

In the past few decades, the impact of human activities on natural ecosystems has led to a significant increase in the extinction rate. According to the Intergovernmental Science-Policy Platform on Biodiversity and Ecosystem Services (IPBES) report, more than 500,000 species have an insufficient habitat for long-term survival due to habitat loss and fragmentation [1]. *Orchidaceae* is one of the largest and most widely distributed families of Angiosperm plants, with more than 28,000 species and 763 genera [2]. During their evolutionary process, orchids have adapted to different environments, and these species are absent only from desert and polar regions [3]. Although orchids occupy a wide range of habitats, several of them are extremely rare. This group of plants is the most threatened by anthropic impact and environmental changes. As a matter of fact, more than half (56.5%) of the only 948 orchid species estimated worldwide using the Global Red List Criteria are considered threatened [4]. Therefore, all orchid species are protected in several countries by regional, national and international legislation, such as the Convention on International Trade in Endangered Species of Wild Fauna and Flora (CITES) [5,6]. Habitat degradation, weed invasion, herbivory, illegal harvesting, pollinator decline, and climate change are the main threats to orchid survival [7]. All these factors can negatively affect the population dynamics and long-term viability of orchids. The flowering and survival of many European orchid species is closely related to the resource status of plants, which is, in turn, influenced by appropriate site management [8]. Moreover, these species are expected to be at a greater risk of extinction as they are dependent on interactions with mycorrhizal fungi and pollinators, which are also being affected by habitat loss and climate change [9,10]. Orchids are particularly sensitive to environmental changes and are often amongst the first plant species to disappear in response to anthropogenic disturbances, making orchids relevant bioindicators of the ecological quality of ecosystems [11,12]. Consequently, orchids pose unique challenges for in situ conservation, which is a focus target for many European countries where habitat fragmentation has led to a decline in habitat connectivity and the reduced fitness of different species [13,14].

For the reasons explained above, establishing an extensive network of protected areas is one of the main tools with which to protect endangered species, habitats, and ecosystems, and to neutralize biodiversity loss [15]. Based on the Habitats Council Directive 92/43/EC, the European Union created the Natura 2000 Network, the largest coordinated multinational network of protected areas in the world, which includes more than 18% of the EU’s land area and more than 8% of its marine territory [16,17]. The role of protected areas in the prevention of species extinction is still uncertain from several cases. Studies have stated that there is a marked difference between the conservation expectations and the actual effectiveness of these areas [18,19]. This is mainly due to the absence of systematic management planning that can furthermore vary under different socio-economic contexts [20]. In Italy and overall in Europe, several rare orchid species are found in dry grasslands, especially the semi-natural dry grasslands habitat listed in Annex I of the Habitats Directive (92/43/EEC) as 6210(*): ‘semi-natural dry grasslands and scrubland facies on calcareous substrates (*Festuco-Brometalia*)’ [21]. Unfortunately, the abandonment of traditional agricultural practices for high-intensity management and urbanization has caused significant ecological and structural changes in these habitats, as the decrease in niche availability and the enhancement of the dominance of a few species with an impact on weak competitors, such as orchids, has led, in general, to a decline in plant species richness [22]. As the Natura 2000 network is directed to guarantee nature protection ‘taking into account economic, social and cultural requirements and regional and local characteristics’, many environments (especially grasslands and scrublands) fall within urban or man-made areas that are considered as typical green spaces. Consequently, a sustainable economic and ecologic management of these spaces is not always appropriately planned, which would take due account of the ecological requirements of the species and environment. In the Po Delta region (Northern Italy), during the period of strong urbanization related to seaside tourism in the 1960s, many natural areas and habitats (e.g., pine forests, wet and dry meadows and grasslands) were destroyed or incorporated as urban green spaces covered by natural vegetation. Unfortunately, these areas have been managed with urban policies using, for example, several mowing phases during the year, because unmowed meadows with tall grass are commonly regarded as synonymous with mess, not cared for or non-aesthetic [23,24]. Over the years, these meadows have been enriched with non-native plants from neighboring gardens, but they still represent the natural habitat of wild plants, including several orchid species. Hence, these meadows, whether they are part of the Natura 2000 network or not, often represent relict or fragmented habitats that must be protected with proper management for biodiversity conservation and environmental sustainability [25,26].

In this study, we monitored the populations of four native orchid species, both in the protected and unprotected areas of the semi-dry grasslands in the Po Delta, which is considered one of the most important natural areas in Europe. However, this territory has been strongly modified by human activities related to urbanization and agricultural development [27]. So, the Po Delta Regional Park Emilia-Romagna, which encompasses several protected areas that aim to conserve biodiversity in a highly anthropic environment and prevent ecosystem degradation and species extinction, was chosen [28]. The objective of this study was to establish the vitality of these orchid populations by monitoring both their vegetative and reproductive traits. Moreover, through the recording of threats to orchid vitality, we compared the protected and unprotected areas to test the effectiveness of actual protections, based on the assumption that the orchids in protected areas would have a higher conservation status.

## 2. Results

### 2.1. Vegetative and Reproductive Traits

There were, overall, modest differences between the protected and non-protected sites in terms of their vegetative traits (Figure 1). *Ophrys sphegodes* Mill. presented with larger rosettes in the protected sites (Figure 1I), and *Anacamptis morio* (L.) R.M. Bateman, Pridgeon et M.W. Chase had taller stems in the protected sites as well (Figure 1B). Conversely, *Anacamptis pyramidalis* (L.) Rich. had a higher SLA and higher LDMC in the non-protected sites (Figure 1N,R). *Anacamptis coriophora* (L.) R.M. Bateman, Pridgeon et M.W. Chase presented with a contrasting pattern of vegetative traits in relation to the protection level, with more leaves in the protected sites, but with taller stems and a higher SLA in the non-protected sites (Figure 1D,H,O).

There were strong differences among the species with respect to their reproductive traits. Indeed, all the species flowered, with only *A. morio* presenting more flowers in the non-protected sites (Table 1). However, both *O. sphegodes* and *A. morio* produced occasional, if any, fruit because the great majority of their flowers were not fecundated, and rotted without setting fruits during ripening in these two species. For this reason, the number of fruits, seed mass and number of embryos could be recorded only for *A. pyramidalis* and *A. coriophora* (Table 1). These two species presented similar patterns for their reproductive traits in relation to the protection level, with an equal performance in the protected vs. non-protected sites for most traits. There were more fruits in the protected sites for *A. pyramidalis*, and heavier seeds as well as more embryos in the protected sites for *A. coriophora* (Table 1).

### 2.2. Vitality of Orchid Individuals and Populations

The vitality of the *O. sphegodes* and *A. morio* individuals were much higher in terms of their vegetative traits than their reproductive traits, with negligible differences related to the protection level (Figure 2A,B). The *A. pyramidalis* individuals also presented with a higher vegetative vitality than reproductive vitality, although with less strong of a difference between the traits (Figure 2C). Conversely, the *A. coriophora* individuals had a higher reproductive vitality than vegetative vitality (Figure 2D). The reproductive vitality of the *A. pyramidalis* individuals was higher in the non-protected sites than in protected sites, while the reverse was true for *A. coriophora* (Figure 2C,D).

The population vitality was higher in terms of the vegetative traits compared to the reproductive traits for *O. sphegodes*, *A. morio* and, although to a lesser extent, for *A. pyramidalis*. The reverse was true for *A. coriophora* (Table 2). For *O. sphegodes* and *A. morio,* the Q values suggested prosperous population conditions in term of vegetative traits, while for *A. pyramidalis* and *A coriophora,* the Q values suggested moderately prosperous to equilibrium conditions in terms of the vegetative traits (Table 2). Both *O. sphegodes* and *A. morio* presented with very low Q values in terms of their reproductive traits, indicating a depressed population condition independent of the protection level. *A. pyramidalis* had somewhat higher reproductive Q values in both the protected and non-protected sites compared to *O. sphegodes* and *A. morio*, indicating equilibrium conditions (Table 2). *A. coriophora* had much higher reproductive Q values in the protected sites, indicating prosperous population conditions, than in the non-protected sites, where the population condition was in equilibrium (Table 2).

### 2.3. Environmental Threats

The highest environmental threat was the invasion of alien plant species (IAS), with differing levels for the other threat types considered (Figure 3). Environmental protection did not imply reduced threat levels, with the exceptions of herbivory and IAS for *A. coriophora* (Figure 3).

## 3. Discussion

In terms of the vegetative traits, both *O. sphegodes* and *A. morio* had a high performance independent of the protection level. The populations of both species were classified as prosperous in both the protected and non-protected areas. In contrast, the vegetative performance of *A. pyramidalis* and *A. coriophora* were somewhat lower. Consequently, the populations of *A. pyramidalis* and *A. coriophora* were classified as moderately prosperous in the non-protected areas and even in equilibrium in the protected areas. Such a rather strong difference in the vegetative performance between *O. sphegodes* and *A. morio*, on the one side, and *A. pyramidalis* and *A. coriophora* on the other side, was related to the different phenology of the four species. Indeed, both *O. sphegodes* and *A. morio* have wintergreen rosettes and aboveground stems that develop early in spring [29,30], while *A. pyramidalis* and *A. coriophora* both lack overwintering rosettes and their aboveground organs start developing more than a month later [31,32]. For this reason, the four species reacted differently to mowing. Although mowing is believed to be advantageous for orchid fitness, its occurrence does not automatically ensure the vitality of orchid populations [33,34,35]. Mowing conducted outside of the growing period has been found to improve the vitality of orchid populations. Early mowing alleviates the competitive pressure from species that can outcompete orchids because of a higher growth potential [36]. Late mowing removes the old plant biomass, reducing the shading of orchids in the subsequent year and increasing the light available for photosynthesis [37,38]. In our study area, mowing generally is practiced intensively from late April to mid-May in order to ameliorate green areas for public use. Because of their early phenology, *O. sphegodes* and *A. morio* were unharmed by the mowing that took place since the growing season for both species had already ended. Conversely, *A. pyramidalis* and *A. coriophora* were more or less heavily damaged by mowing during their growing season. In particular, many individuals of *A. pyramidalis* were wiped out at the most strongly mowed sites. To our knowledge, there are no well-defined prescriptions for regulating the mowing timing in the Po Delta Park. Hence, the environmental protection in our study area was unsuccessful at preventing damage to the orchid populations because of mowing during the inappropriate season. Although there were, overall, few differences in the vegetative traits between the protected and non-protected areas for both *A. pyramidalis* and *A. coriophora*, the slightly lower vegetative performance in the protected areas was prevalently associated with a lower SLA. As the SLA represents a powerful proxy for the photosynthetic capacity of vascular plants [39,40], a reduced SLA may constrain photosynthetic activity, ultimately limiting plant growth. A smaller plant size may be detrimental to vitality because medium-size and big plants contribute more to the persistence of orchid populations [41]. High photosynthetic rates can also improve the reproductive success of orchid species [42,43]. We have no mechanistic explanation as to why the vegetative vitality of *A. pyramidalis* and *A. coriophora* were, albeit slightly, worse in the protected areas. However, it has to be considered that the small-scale environmental variations associated, for example, with mosaic-like patterns of canopy height can influence the vegetative performance of orchids independent of protection status [44,45].

The reproductive success of *O. sphegodes* and *A. morio* was practically nil, in spite of their high vegetative performance. So, the populations of both species were classified as depressed in terms of their reproductive traits. The lack of reproductive success in *O. sphegodes* and *A. morio* was totally unrelated to the threats detected in our surveys and to the protection level as well. Failed sexual reproduction in some orchids species has been linked to adverse weather conditions in the previous growing season. For example, Kirillova and Kirillov [46] observed less flowers and smaller seeds in *Platanthera bifolia* the year after a dry summer season. Although our study area did experience extremely hot, dry weather during summer 2022, viz., the year preceding our sampling (https://www.arpae.it/it/temi-ambientali/clima/clima (accessed on 22 December 2023)), it is very unlikely that this was the cause hampering sexual reproduction in *O. sphegodes* and *A. morio* in 2023. Indeed, a summer drought causes no problems for species which produce leaves in the autumn and remain green as they have no above-ground organs [47]. The observed failure of sexual reproduction in *O. sphegodes* and *A. morio* was determined by the lack of fruit setting, even if both species flowered vigorously. We did not observe any apparent sign of pest occurrence on the stems of these orchids. Therefore, we believe that failed pollination was the most likely cause of lacking fructification in *O. sphegodes* and *A. morio*, for example, due to a temporal displacement in the phenology of the plant and pollinator [48]. Indeed, both *O. sphegodes* [29] and *A. morio* [49] require a very narrow specialized set of pollinators to be available for successful pollination. Strictly deceptive orchid species, like *O. sphegodes* and *A. morio*, usually flower early and have lower fructification rates, with even <15% of the flowers producing capsules [50,51,52]. Thus, their strategy seems to be based on a low-risk approach that ensures the production of a smaller number of fruits before the shading effect from the surrounding plant community and the potential impact of mowing occurs [37]. However, if sexual reproduction fails for several consecutive years, this could considerably reduce the population size because vegetative multiplication occurs rather occasionally through the survival or splitting of the old tuber [53].

Both *A. pyramidalis* and *A. coriophora* set fruit containing viable seeds that produced embryos, even if the reproductive vitality was overall higher in *A. coriophora*. This finding is consistent with the pollination strategy of the two species, with *A. pyramidalis* possessing a generalized deceptive behavior because their flowers contain nectar in the spur [54], while *A. coriophora* acts as a fully rewarding species [32]. All types of threat recorded in our survey besides mowing, viz. waste dumping, herbivory and IAS, could to some extent lower the reproductive vitality of *A. pyramidalis* and *A. coriophora,* although the populations of both species never attained a depressed condition in terms of their reproductive traits. In addition to the direct and indirect effects of waste dumping on environmental pollution [55], waste dumping can bring about habitat loss and fragmentation, both of which represent major threats to the viability of orchid populations [56]. Our data do not allow us to evaluate whether and to what extent waste dumping impacted the vitality of the orchid populations. However, as the objective of our study was to assess the effectiveness of environmental protection for ensuring the vitality of orchid populations, we can state that the threat associated with waste dumping was unrelated to the protection level for any of the four species investigated. In our study area, herbivory mainly consists of browsing by fallow deer (*Dama dama*). While grazing practices by sheep and cattle have been shown to contribute to the preservation of orchids in semi-dry grasslands, not only by appropriate biomass removal but also by selective defoliation and the creation of gaps suitable for orchid seed germination [57,58], browsing by deer has been shown to have detrimental effects on several plant species [59]. For example, browsing by the North American white-tailed deer (*Odocoileus virginianus*) negatively impacts the vegetative growth of the rare orchid *Platanthera integrilabia* [60], which, in turn, hampers the reproductive success of the species [42,43]. Browsing by white-tailed deer has also been found to hamper the population viability of the herb *Panax quinquefolius*, through the removal of foliage, flowers and fruits [61]. In general, herbivory has negative effects on floral traits, plant attractiveness to pollinators and, eventually, on a plant’s reproductive success. The damage can either derive from the direct effects of browsing on the reproductive organs, or by the indirect effects on reproduction determined by defoliation, which can lead to decreased seed production, reduced leaf growth and flowering in subsequent years [62,63]. Moreover, intensive herbivory by ungulates can enhance the invasion of alien species because many invasive plants have adapted to habitat disturbance, so that their survival is favored as ungulates eliminate the native plants, which are more palatable than the invasive species [64]. In addition to possible interactions with herbivory [65], IAS alters ecological interactions since the invasive species are superior competitors for light and space compared to the native species [66]. Wild orchids often tolerate IAS, but their survival is negatively impacted once the cover of alien species exceeds 20% [67]. Invasive alien species also hamper the reproductive success of native species by reducing the population fitness through reduced seed sets, germination rates and seedling establishment [68]. So, in addition to outcompeting native orchids thanks to their superior competitive ability, IAS can exert a negative impact on orchid reproduction, for example, by producing novel chemical compounds [69] or by reducing the diversity of mycorrhizal fungi [70,71]. Environmental protection was effective at lowering the threat brought about by herbivory and IAS for sites hosting *A. coriophora* populations, but not for the sites hosting *A. pyramidalis* populations. This was because the *A. coriophora* sites were located in areas less frequented by fallow deer. In these areas, environmental protection was furthermore able to act as a filter against IAS [72,73]. The population vitality of *A. pyramidalis* was, unexpectedly, lower in the protected than in the non-protected areas. Such a seemingly paradoxical finding may be due to uninvestigated micro-environmental factors that negatively affected the reproduction of *A. pyramidalis* in the protected sites [74,75].

## 4. Materials and Methods

### 4.1. Study Area

This study’s area is located in the north-easternmost part of the region Emilia-Romagna (Northern Italy), along the North Adriatic coast (about 44°32′–44°56′ N; 12°08′–12°16′ E; Figure 4). Most of this area is within the territory of the Po Delta Regional Park Emilia-Romagna. However, the park does not cover a continuous area but is divided into six parts (locally called stations) separated from each other by non-protected areas. The Regional Park has a total surface area of 54,000 ha and hosts sixteen sites within the Natura 2000 network, either sites of community importance or special areas of conservation (https://www.parks.it/parco.delta.po.er/Epar.php (accessed on 21 December 2023)). In addition, the northern part of this area hosts the National Nature Reserve Bosco della Mesola. According to the Köppen classification of climates, the climate in this study area is temperate continental, with a mean annual temperature of about 14 °C. The mean temperature of the coldest month (January) is about 3 °C and the mean annual temperature of the warmest month (July) is about 24 °C. The mean total annual precipitation is 600–700 mm, with a short period of weak summer aridity [76].

### 4.2. Species Selection, Sampling and Laboratory Analyses

We have been investigating wild orchids in the Po Delta Park since 2012. Fifteen species of native orchids are currently present in this study area [77]. We selected for sampling the orchid species occurring in at least three protected sites and three non-protected sites in order to ensure the presence of replicated samples. Four species met these requisites: *Ophrys sphegodes* Mill.; *Anacamptis morio* (L.) R.M. Bateman, Pridgeon et M.W. Chase; *Anacamptis pyramidalis* (L.) Rich.; and *Anacamptis coriophora* (L.) R.M. Bateman, Pridgeon et M.W. Chase. These species occur in semi-dry grasslands rich in annual species, such as *Silene conica*, *Cerastium semidecandrum*, *Vulpia membranacea* and *Medicago minima*. These grasslands lie on sandy flat areas corresponding to the ancient dunes leveled by atmospheric agents. Syntaxonomically, these grasslands are most similar to communities of *Sileno conicae–Cerastietum semidecandri* and can, therefore, be included in the priority habitat 2130 (fixed coastal dunes with herbaceous vegetation, ‘grey dunes’). The presence of some representative perennial species, such as *Sanguisorba minor*, *Scabiosa columbaria*, *Fumana procumbens* and *Helianthemum apenninum*, besides the richness of the orchid species, presents some resemblance to habitat 6210 (http://vnr.unipg.it/habitat/collaboratori.jsp (accessed on 21 December 2023)). The soil in these grasslands is sandy, subneutral, with a low humus content (total soil carbon concentration < 10 mg g^−1^) and rather poor in nutrients. There were no evident differences in terms of soil chemistry between the protected and non-protected sites (Appendix A).

During the years 2021–2022, we surveyed several sites both in the protected and non-protected areas in order to define the phenological phases of the four selected species. We also counted the number of flowering individuals in a number of 1 × 1 m quadrats, with the objective of perceiving the interannual fluctuations in the numerical size of the orchid populations. In March 2023, we selected 15 sites where the four species had the highest abundance for detailed measurements of the vegetative and reproductive traits. Seven of these sites were in protected areas and eight were in non-protected areas (Figure 1; Appendix A). All the sites were located in open areas far enough away from trees or buildings that could cast shadows on the vegetation. The light level during the central hours of clear days was never <800 µmol PPFD m^−2^ s^−1^. We visited the sites several times during spring 2023 depending on the phenology of the four species (Figure 5). For each species, at the peak of the flowering period we randomly chose 30 sound individuals in three protected sites and three non-protected sites. Only for *A. coriophora* could we locate an additional site in a non-protected area. In a minority of cases, the site hosted a small orchid population with fewer than thirty individuals. In those cases, all of the individuals occurring at the site were sampled. The individuals selected were tagged by means of a numbered wooden stick inserted into the ground about 1.5 cm from of the stem. The height of the stem and the diameter of the basal rosette were measured in the field with a ruler, and the number of flowers was counted on each individual. We also counted the number of leaves on each selected individual. One healthy rosette leaf for each individual was harvested and carried to the laboratory for subsequent analyses. For each species, at the peak of fruit ripening we counted the number of fruits present on each of the tagged individuals. Afterwards, all the fruits on a given individual were harvested, sealed in a plastic bag wrapped with wet paper to keep them hydrated and carried to the laboratory for subsequent analyses.

The day after, the leaves were weighed and scanned (CanoScan LiDE 120, Canon Italia, Cernusco sul Naviglio (MI), Italy) in order to determine the fresh leaf mass and leaf area. The same leaves were weighed again after being dried at 105 °C for 72 h in order to determine the leaf dry mass. All the harvested fruits were carefully emptied of seeds. The latter were pooled on a paper sheet and weighed. A subsample of 100 seeds was picked from the pool and examined under a stereo microscope to count the embryos.

### 4.3. Data Compilation and Statistical Analyses

The data obtained by field measurements and subsequent laboratory analyses were arranged into two groups:Vegetative traits: Stem height (measured in the field); number of leaves (counted in the field); rosette diameter (measured in the field); specific leaf area (SLA, the ratio between the leaf area and leaf fresh mass as obtained by laboratory scanning and weighing); and leaf dry matter content (LDMC, the ratio between the dry mass and fresh mass as obtained by laboratory weighing).Reproductive traits: Number of flowers (counted in the field); number of fruits (counted in the field); seed mass (as obtained by laboratory weighing); and number of embryos (counted in the laboratory under a stereo microscope).

The four species differed greatly from each other in terms of all the vegetative and reproductive traits because of the intrinsic differences resulting from their evolutionary legacy. Hence, it would be meaningless to assess the interspecific differences based on the rough data. We, therefore, ran separate one-way ANOVAs with the protection level of the sampling sites (protected vs. non-protected) as the factorial factor, for all the traits of the four species.

For comparing the vitality of the orchid individuals, in terms of both the vegetative and reproductive traits, we calculated a normalized index (Q_individual_) based on fractions of the maximum values [78]. The normalization was performed as in Formula (1)
Q_individual_ = X_i_/X_max_
(1)
where X_i_ is the rough value of a given trait for an individual of a given orchid species in a given site, and X_max_ is the maximum rough value across the whole set of the above values. The Q_individual_ values were then pooled separately for the vegetative traits and reproductive traits. Toward this end, we calculated the mean values of Q_individual_ for the five vegetative traits and the four reproductive traits, respectively, for all the individuals of a given orchid species in a given site. These mean values were statistically analyzed by two-way ANOVAs with the trait (vegetative vs. reproductive), protection level (protected vs. non-protected) and their interaction as the factorial factors. The significance of differences among the means was assessed by Fisher’s LSD post hoc tests at a *p*-level < 0.05.

The vitality of the orchid populations in the protected vs. non-protected sites was assessed by splitting the mean Q_individual_ values for the vegetative and reproductive traits into three categories, as follows: [27] vitality class a: mean Q_individual_ > 0.66; vitality class b: 0.33 > mean Q_individual_ < 0.66; and vitality class c: mean Q_individual_ < 0.33. The overall vitality of the orchid populations was eventually estimated by calculating the fraction of individuals within classes a and b, according to Formula (2):Q = (a + b)/2 (2)

The Q values, in the range from 0 to 0.5, were classified as follows: prosperous: Q > 0.333; equilibrium: 0.167 > Q < 0.333; and depressed: Q < 0.167.

Whenever we visited the sites, we assessed visually the level of four types of potential or real risks to the vitality of the orchid populations, viz., mowing, waste dumping (waste), browsing by herbivores (herbivory) and invasion of alien species (IAS). At the end of the season, we summarized the levels of the four risk types for each site using four categories: 1: absent; 2: weak; 3: moderate; and 4: high.

## 5. Conclusions

We conclude that environmental protection per se is unlikely to effectively preserve endangered plant species, in particular, the orchids in our study area, echoing the findings of previous studies [79,80]. Targeted actions against specific environmental threats and targeted actions translated into explicit prescriptions in the regulations for the protected areas must be defined to achieve the effective protection of orchids.

## Figures and Tables

**Figure 1 plants-13-00610-f001:**
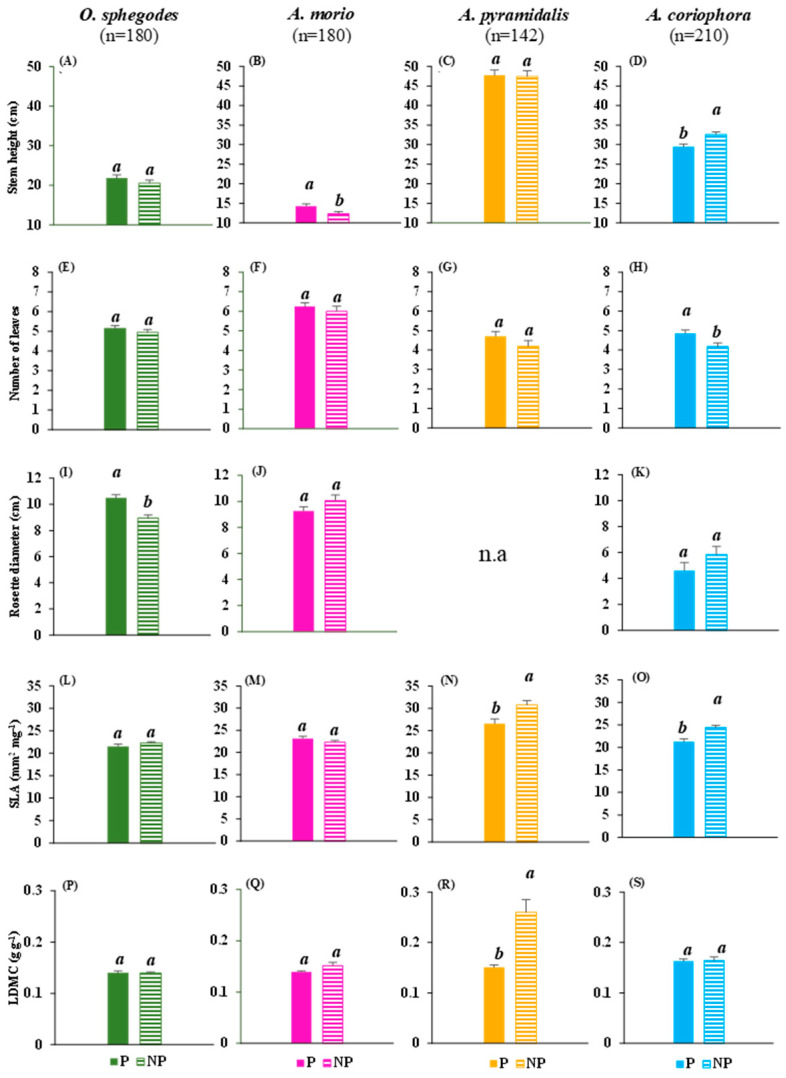
The mean (+1 SE) values of the vegetative traits. (**A**–**D**): stem height; (**E**–**H**): number of leaves; (**I**–**K**): rosette diameter; (**L**–**O**): specific leaf area, SLA; (**P**–**S**): leaf dry matter content (LDMC) for the four orchid species at the protected (P) and non-protected (NP) sites. Within each panel, the means followed by the same letter do differ significantly (*p* < 0.05) between the P and NP, obtained based on Fisher’s LSD post hoc tests; n.a. = not available; n = number of samples.

**Figure 2 plants-13-00610-f002:**
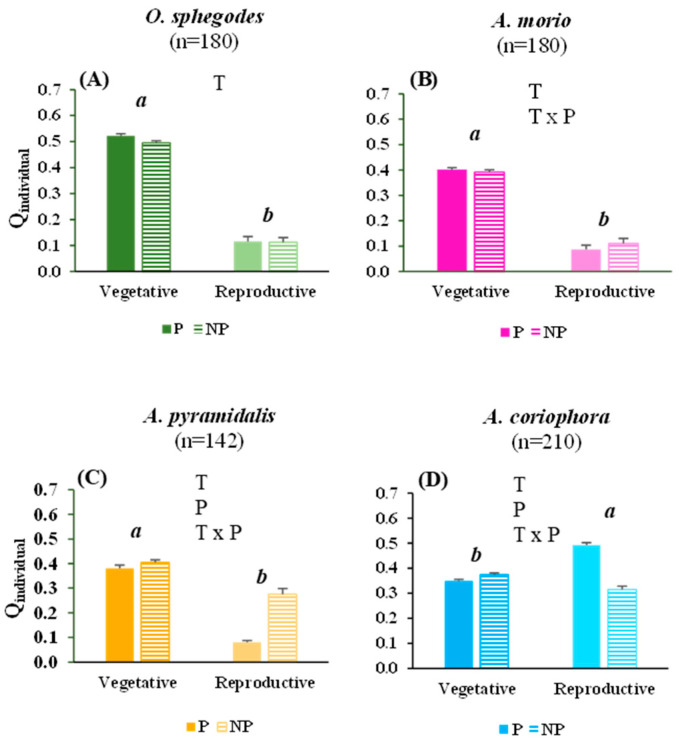
The mean (+1 SE) values of Q_individual_ for the vegetative and reproductive traits of the four orchid species (**A**–**D**) at the protected (P) and non-protected (NP) sites. The significant (*p* < 0.05) effects of trait (T), protection level (P) and their interaction, obtained by two-way ANOVAs, are indicated by the letters within each panel. Different letters indicate significant (*p* < 0.05) differences between the protected and non-protected sites for any combination of trait and protection level; n = number of samples.

**Figure 3 plants-13-00610-f003:**
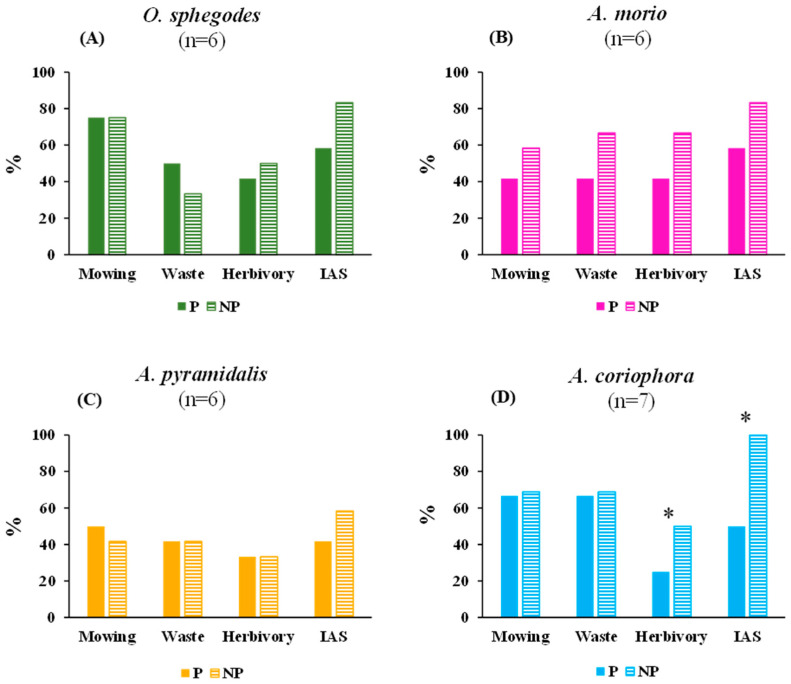
The mean values of four types of environmental risks (mowing; waste dumping, waste; browsing by herbivores, herbivory; invasion of alien species, IAS) to the four orchid species (**A**–**D**) at the protected (P) and non-protected (NP) sites. The asterisks indicate significant (*p* < 0.05) differences between the protected and non-protected sites obtained by one-way ANOVAs. For clarity of presentation, the values are percentages of the maximum value (for details of the raw data and ANOVA results see Appendix A); n = number of samples.

**Figure 4 plants-13-00610-f004:**
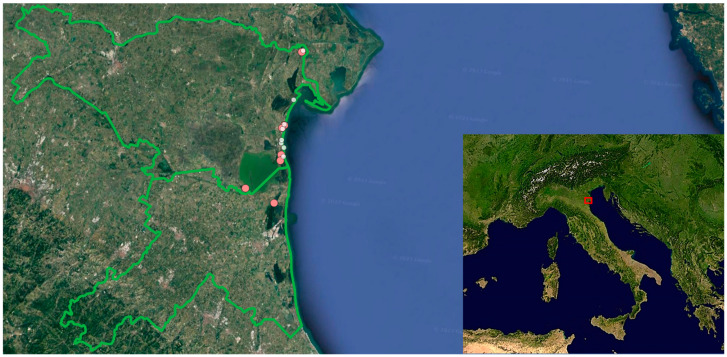
A map of this study area. The larger pink dots are the sampling sites in the protected areas; the smaller white smaller are the sampling sites in the non-protected sites.

**Figure 5 plants-13-00610-f005:**
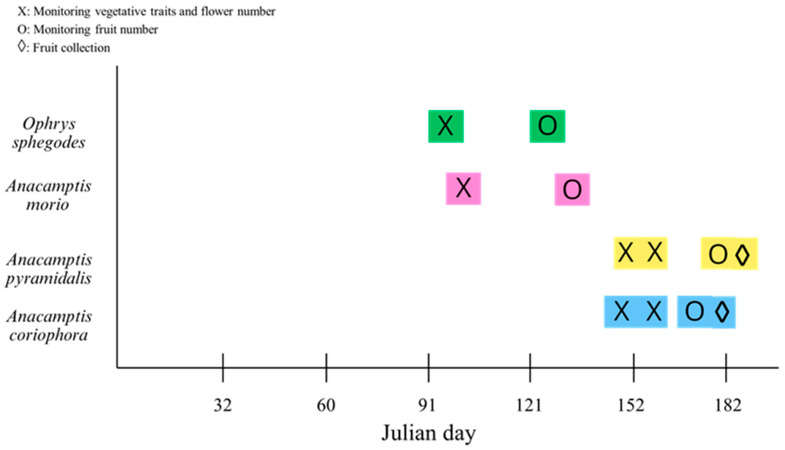
Timing of the field sampling dates. The sampling dates of the four species were highlighted by the same colors used in Figure 1.

**Table 1 plants-13-00610-t001:** The mean (±1 SE) values of the reproductive traits of the four orchid species at the protected (P) and non-protected (NP) sites. Within each group, the means followed by the same letter differ significantly (*p* < 0.05) between the P and NP, obtained based on Fisher’s LSD post hoc tests; n.a. = not available.

Trait		*O. sphegodes* (n = 180)	*A. morio* (n = 180)	*A. pyramidalis* (n = 142)	*A. coriophora* (n = 210)
Number of flowers	P	4.60 ± 0.20 a	7.61 ± 0.37 b	44.20 ± 2.97 a	22.89 ± 0.79 a
	NP	4.48 ± 0.19 a	9.66 ± 0.46 a	49.47 ± 2.78 a	20.93 ± 0.74 a
Number of fruits	P	n.a.	n.a.	2.79 ± 0.42 a	15.80 ± 0.70 a
	NP	n.a.	n.a.	1.36 ± 0.22 b	13.86 ± 0.91 a
Seed mass (mg)	P	n.a.	n.a.	0.78 ± 0.12 a	5.30 ± 0.30 a
	NP	n.a.	n.a.	0.69 ± 0.11 a	2.00 ± 0.20 b
Number of embryos	P	n.a.	n.a.	42.6 ± 6.0 a	96.37 ± 1.90 a
	NP	n.a.	n.a.	39.8 ± 5.4 a	48.45 ± 4.52 b

**Table 2 plants-13-00610-t002:** The Q values indicating population vitality of the four orchid species in terms of the vegetative (V) and reproductive (R) traits at the protected (P) and non-protected (NP) sites.

Species	Protection	Trait	Q
*O. sphegodes* (n = 180)	P	V	0.494
	NP	V	0.500
	P	R	0.079
	NP	R	0.082
*A. morio* (n = 180)	P	V	0.428
	NP	V	0.383
	P	R	0.058
	NP	R	0.076
*A. pyramidalis* (n = 142)	P	V	0.303
	NP	V	0.395
	P	R	0.220
	NP	R	0.211
*A. coriophora* (n = 210)	P	V	0.311
	NP	V	0.350
	P	R	0.483
	NP	R	0.267

## Data Availability

The data presented in this study are available upon request from the corresponding author. The data are not publicly available due to privacy.

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
