# Peer review of "How Effective Is Environmental Protection for Ensuring the Vitality of Wild Orchid Species? A Case Study of a Protected Area in Italy"

_plants, 2024, doi:10.3390/plants13050610_

Round 1

Reviewer 1 Report

Comments and Suggestions for Authors

The manuscript submitted for review presents a interesting issue however a few queries, comments:

1. With what test, at what level of significance were statistical calculations performed?

2. Observations were carried out in only one year? No information given.

3. What differentiates this area of observation? Is there any soil variability? The observation areas are very similar to each other. One year observations in my opinion is not enough. There should be observations of at least 2 years and in the current manuscript it is not even known in which year the surveys were performed. If it is the case that these are 1-year observations, the manuscript rather to  reject. I look forward to the clarification.

4. No chapter – Conclusions

5. Font in a different format from the other chapters of the manuscript - please refer to the requirements of the journal in terms of literature formatting

Author Response

REVIEWER #1

  1. With what test, at what level of significance were statistical calculations performed?

See clarification on l. 330-331.

  1. Observations were carried out in only one year? No information given.

We have surveyed orchid populations in this area since 2012. We counted the flowering individuals of the selected species in 2021-2022. Detailed measurements of vegetative and reproductive traits were performed in 2023. The protocol is now described in more detail (see l. 278-284).

  1. What differentiates this area of observation? Is there any soil variability? The observation areas are very similar to each other. One year observations in my opinion is not enough. There should be observations of at least 2 years and in the current manuscript it is not even known in which year the surveys were performed. If it is the case that these are 1-year observations, the manuscript rather to reject. I look forward to the clarification.

Detailed soil analysis was outside the scope of our study. However, we provided some preliminary data on soil chemistry which does not show evident differences in soil chemistry either among species or between protection level (l. 274-277 and Supplementary Table S1).

  1. No chapter – Conclusions

We rearranged the last paragraph of the Discussion and created a short Conclusion paragraph (l. 345-350).

  1. Font in a different format from the other chapters of the manuscript - please refer to the requirements of the journal in terms of literature formatting

I guess that the font in the References paragraph is adapted during the editing phase.

Reviewer 2 Report

Comments and Suggestions for Authors

The Authors present an interesting topic in a well-written manuscript. However, several questions and doubts arise during exploration.

The major concern is the plants of different populations, conditions, ages, etc., used in the study and their comparability. Actually, similarly (pre-)cultivated plants should be planted in various habitats for further comparison.

Also, one wonders why exactly these four plant species but no other species were selected? Which specialty predestined the chosen species?

It is suggestive to describe the species‘ natural growth requirements in detail, including pollination particularities, such as pollinators, seasonalities, remontant and dormancy properties, etc.
Further, the sampling sites need a more detailed description per each; showing qualitative photographs of each location is suggestive. – SLA depends a lot, among other things, on light conditions, which could have been quite variable in the different habitats.

At the present state, one cannot follow the sampling procedure and repeated samplings; a schematic workflow including a timeline of repeated samplings is highly suggestive. Taking out plant parts (rosettes) for sampling seems critical for overall plant fitness.

The study-specific evaluation of environmental risks and their pooling with plant trait values is unclear and needs a much more detailed description.

Figure legends & Table headings should explain which data have been pooled for the presented graphs and should be mentioned in detail; n = ?

Is this the correct MDPI Plants style of legends and headings? - Scientifically correct: Figures have subscriptions, and Tables have headings.

Finally, based on the limited data obtained in a single region, one may draw general conclusions for global environmental protection areas.

Perhaps the study could be published by changing the title, modifying the goal and conclusions, and providing more details for plant biology/physiology/growth and concrete habitat conditions.

D E T A I L E D     C O M M E N T S

Write the full Latin species names, also providing the family name, in 2.1., because they are mentioned for the first time in the manuscript in this subchapter.

Figure numbering requires revision:
Figure 2
Figure 1
Figure 3
Figure 2
Figure 4
Figure 3
Figure 1
Figure 4

Results
Figure 2
>> The asterisks should be shifted between the columns instead of placing them on one column.

> 2.2. Vitality of orchid individuals and populations

> 2.3. Environmental threats

Discussion
> We conclude that environmental protection per sé is unlikely to effectively preserve endangered plant species [76], in particular orchids [77], in the north-easternmost part of the region Emilia-Romagna (Northern Italy) within the territory of the Po Delta Regional Park Emilia-Romagna.
>> One cannot conclude from the present study to all environmentally protected areas worldwide!

MM
4.1.
> but is divided into six regions
> mean annual temperature of about 14 °C
Figure 1
Figure 4. Map of the study area.
>> Is this Figure available at a higher resolution?
>> Wouldn´t it make sense to number and define each sampling site and surrounding green areas to refer to the text and vice versa and assign the plant species to the sampling sites?

4.2.
>> Why the leaves have been rehydrated? Couldn´t they be kept hydrated after sampling to avoid any side effects?
> in order to determine fresh leaf mass and leaf area
> in order to determine the leaf dry mass
>> Why the leaves have been dried at 105 °C? Such a high temperature leads to leaf denaturation and not representative measurements of leaf mass.
> invasion of alien plant and animal species (IAS) >>> or different?

Author Response

REVIEWER #2

Comments and Suggestions for Authors

The Authors present an interesting topic in a well-written manuscript. However, several questions and doubts arise during exploration.

The major concern is the plants of different populations, conditions, ages, etc., used in the study and their comparability. Actually, similarly (pre-)cultivated plants should be planted in various habitats for further comparison.

We agree that planting pre-cultivated plants in natural and/or controlled habitats can fruitfully add to our knowledge of orchid vitality. However, this was not the very objective of the present study.

Also, one wonders why exactly these four plant species but no other species were selected? Which specialty predestined the chosen species?

The criteria of species selection are now stated on l. 261-264.

It is suggestive to describe the species‘ natural growth requirements in detail, including pollination particularities, such as pollinators, seasonalities, remontant and dormancy properties, etc.

We recognize that lack of data on pollinators may be a weak point of our study. Unfortunately, this field of investigation is beyond our expertise so we just referred to literature data as regards to pollination habit of the four target species.

Further, the sampling sites need a more detailed description per each; showing qualitative photographs of each location is suggestive. – SLA depends a lot, among other things, on light conditions, which could have been quite variable in the different habitats.

We agree. So, we provided some data about light level (l. 284-287) and soil chemistry (Supplementary Table S1). In addition, we provided photos of selected sites in Supplementary Figure S1.

At the present state, one cannot follow the sampling procedure and repeated samplings; a schematic workflow including a timeline of repeated samplings is highly suggestive. Taking out plant parts (rosettes) for sampling seems critical for overall plant fitness.

We described in more detail the sampling protocol also with support of the schematic workflow of the sampling timeline in Figure 5.

The study-specific evaluation of environmental risks and their pooling with plant trait values is unclear and needs a much more detailed description.

Regrettably, no quantitative data was available about the level of damage experienced by our sites. So, we could only assess environmental threats by visual inspection

Figure legends & Table headings should explain which data have been pooled for the presented graphs and should be mentioned in detail; n = ?

See n. within the body of Figures 1-3 and Table 1

Is this the correct MDPI Plants style of legends and headings? - Scientifically correct: Figures have subscriptions, and Tables have headings.

Ok, done.

Finally, based on the limited data obtained in a single region, one may draw general conclusions for global environmental protection areas.

Perhaps the study could be published by changing the title, modifying the goal and conclusions, and providing more details for plant biology/physiology/growth and concrete habitat conditions.

Yes, we realize that it was a bit pretentious to draw general conclusions from a local study. So, we changed the title and attenuated some general sentences by stressing that our comments and conclusions are centred on our case study (see l. 2, 23-24, 345-350).

D E T A I L E D     C O M M E N T S

Write the full Latin species names, also providing the family name, in 2.1., because they are mentioned for the first time in the manuscript in this subchapter.

Ok, done

Figure numbering requires revision:

Figure 2 → Figure 1

Figure 3 → Figure 2

Figure 4 → Figure 3

Figure 1 → Figure 4

Ok, the figures were re-numbered after inserting the new Figure 5 with the workflow.

Results

Figure 2

>> The asterisks should be shifted between the columns instead of placing them on one column.

Ok, done

> 2.2. Vitality of orchid individuals and populations

There were two different elaborations, based on individuals and populations, respectively.

> 2.3. Environmental threats

Ok, done

Discussion

> We conclude that environmental protection per sé is unlikely to effectively preserve endangered plant species [76], in particular orchids [77], in the north-easternmost part of the region Emilia-Romagna (Northern Italy) within the territory of the Po Delta Regional Park Emilia-Romagna.

>> One cannot conclude from the present study to all environmentally protected areas worldwide!

Ok, see the response to the general comments above, in particular l 345-350.

MM

4.1.

> but is divided into six regions

We call these sectors ‘parts’ because the park does not cover a continuous area.

> mean annual temperature of about 14 °C

Ok, done

Figure 1 → Figure 4. Map of the study area.

Ok, re-numbered.

3Is this Figure available at a higher resolution?

Yes, see please the new version of Figure 4.

>> Wouldn´t it make sense to number and define each sampling site and surrounding green areas to refer to the text and vice versa and assign the plant species to the sampling sites?

The sampling sites are overall quite close to each other. In addition, such detail is of no importance for a reader, except perhaps local people.

4.2.

>> Why the leaves have been rehydrated? Couldn´t they be kept hydrated after sampling to avoid any side effects?

This was a trivial error due to the ‘copy and paste’. See l. 298-299.

> in order to determine fresh leaf mass and leaf area

Ok, done

> in order to determine the leaf dry mass

Ok, done

>> Why the leaves have been dried at 105 °C? Such a high temperature leads to leaf denaturation and not representative measurements of leaf mass.

In previous studies we noticed that small amounts of water can remain within leaves if dried at 70°C. As we did not need sound tissues (as happens, for example, when foliar nutrient contents are to be determined), we decided to dry them at 105 °C

> invasion of alien plant and animal species (IAS) >>> or different?

Invasion of alien plant species (see l. 136).

Reviewer 3 Report

Comments and Suggestions for Authors

After careful evaluation of the manuscript title “How effective is environmental protection in ensuring the vitality of wild orchid species?”, you can find below my comments:

- The manuscript must be supplemented with significant analyzes of the chemical properties of the soil in order to be further processed in the journal Plants;

- The purpose of the manuscript is clearly specified, but it is impossible to achieve if the work is not enriched with an analysis of the basic chemical properties of the soil from the places where orchids occur.

I would like to propose the following important amendments:

Results

- For easier interpretation of figures 1-4 and table 1, please insert letter markings (post-hoc analysis);

- Please supplement the results with basic analyzes of the chemical properties of the soil, e.g. soil reaction, humus content, sorption capacity, calcium carbonate content. These parameters will significantly facilitate the interpretation of the results, please complete at least some of them.

Conclusions

- Please add a synthetic summary of your work.

Best regards

Author Response

REVIEWER #3

Comments and Suggestions for Authors

After careful evaluation of the manuscript title “How effective is environmental protection in ensuring the vitality of wild orchid species?”, you can find below my comments:

- The manuscript must be supplemented with significant analyzes of the chemical properties of the soil in order to be further processed in the journal Plants;

- The purpose of the manuscript is clearly specified, but it is impossible to achieve if the work is not enriched with an analysis of the basic chemical properties of the soil from the places where orchids occur.

I would like to propose the following important amendments:

Results

- For easier interpretation of figures 1-4 and table 1, please insert letter markings (post-hoc analysis);

Ok, done

- Please supplement the results with basic analyzes of the chemical properties of the soil, e.g. soil reaction, humus content, sorption capacity, calcium carbonate content. These parameters will significantly facilitate the interpretation of the results, please complete at least some of them.

Detailed analyses of soil features were outside the objective of this paper. Nonetheless, we are aware that better knowledge of some chemical properties of the soil can be useful to support this kind of studies. As we do have some preliminary data on soil chemistry (that will will be explored in more detail in the near future) we provide them in Supplementary Table S1, see also l. 274-277.

Conclusions

- Please add a synthetic summary of your work.

We summarized the results in a short Conclusion paragraph (l. 345-350).

Best regards

Round 2

Reviewer 1 Report

Comments and Suggestions for Authors

The authors corrected the recommended corrections and added the information they were asked to provide. The article can be submitted for publication, however, 2 comments, but technical: 1. Headings – 2.2 and 2.3 – are offset, plus a dot is before the heading 2. Chapter 4.2 - the cited website appears to be written in a larger font

Reviewer 3 Report

Comments and Suggestions for Authors

After careful evaluation of the revised manuscript How effective is environmental protection in ensuring the vitality of wild orchid species?”  I believe that in this form it corresponds to an article that could be published in the Journal Plants. The improved manuscrypt already has greater potential than in its previous form. In the future, please pay more attention to presenting research in a comprehensive form. 

Best regards